# The Efficacy of Ultrasonic Pest Repellent Devices against the Australian Paralysis Tick, *Ixodes holocyclus* (Acari: Ixodidae)

**DOI:** 10.3390/insects12050400

**Published:** 2021-04-30

**Authors:** Amonrat Panthawong, Stephen L. Doggett, Theeraphap Chareonviriyaphap

**Affiliations:** 1Department of Entomology, Faculty of Agriculture, Kasetsart University, Bangkok 10900, Thailand; aor_bio@hotmail.com; 2Department of Medical Entomology, NSW Health Pathology-ICPMR, Westmead Hospital, Westmead, NSW 2145, Australia; Stephen.Doggett@health.nsw.gov.au

**Keywords:** efficacy testing, non-chemical control, tick bite prevention, tick repellent, ultrasonic repellers

## Abstract

**Simple Summary:**

Ultrasonic repellers are widely available and marketed to protect against tick bite. To date, there has been no research on the effectiveness of ultrasonic devices against the Australian paralysis tick, *Ixodes holocyclus*. Thus, this study tested the effectiveness of nine ultrasonic devices with different sound frequencies against female *I. holocyclus*. Testing found that ultrasonic devices produced less than 19.5% repellency. The low-level repellency from ultrasonic repellers means that they cannot be recommended for prevention against tick bite.

**Abstract:**

Ultrasonic pest repellers are often promoted as a means of protecting people and pets from the bites of hematophagous arthropods, such as ticks. However, to date, there has been no published research on the effectiveness of these devices against the Australian paralysis tick, *Ixodes holocyclus* Neumann. The purpose of this study was to test the effectiveness of nine ultrasonic devices against female *I. holocyclus*. Two arenas were constructed, one for the test (with the ultrasonic device) and one for the control (no device). Each arena had a test and an escape chamber, connected by a corridor. Twenty ticks were placed in each test chamber. After the ultrasonic device was operated for 1 h, the number of ticks in both chambers was recorded. Ten replicates were conducted for each device. The average number of ticks that moved from the test to the escape chamber was greater in all the test arenas, with three devices being statistically different from the control. However, the highest percent of ticks that escaped was only 19.5%. This amount is insufficient to offer adequate protection against tick bites and this study adds further weight to previous investigations that ultrasonic devices should not be employed in pest management.

## 1. Introduction

Worldwide, ticks are important vectors in the transmission of a range of pathogenic microorganisms, including protozoa, rickettsiae, bacteria and viruses, to their host animals and to humans [1,2]. In Australia, *Ixodes holocyclus*, commonly known as the Australian paralysis tick, is the most common species that bite humans and is the main species responsible for inducing tick-related morbidity in humans and pets [3,4]. Not only is the species capable of transmitting rickettsia such as *Rickettsia australis* Philip (etiological agent of Queensland tick typhus) [3], the bite of the tick can cause potentially life-threatening toxicosis, tick paralysis, and induce a range of allergic reactions, such as α-Gal syndrome (mammalian meat allergy) [5,6]. With the controversy surrounding the existence (but unproven) presence of Lyme disease in Australia [7,8], much of the contemporary research on *I. holocyclus* has focused on potential pathogens the tick may carry. Indeed, a range of bacteria [9] and viruses have been identified from *I. holocyclus* [10,11], although none of the microbes have yet been linked to human disease.

To prevent these adverse health effects, it is necessary to avoid being bitten by the tick and the main method of bite prevention is through personal protection methods, such as the application of topical repellents applied to clothing and skin [12]. However, up until recently, research into personal protection methodologies against *I. holocyclus* has been totally neglected with not one single publication prior to 2019 [13]. Fortunately, Sukkanon et al. recently tested a range of topical and spatial repellents and made recommendations on which were the most effective products at repelling *I. holocyclus* [13]. Even more recently, Panthawong et al. investigated the use of permethrin-impregnated clothing for repelling *I. holocyclus* [14].

Other methods of preventing *I. holocyclus* bites have yet to be explored in any scientific manner and one of the most controversial of these is the use of ultrasonic devices. Not only are such devices widely available through online retailers, they are commonly sold in Australian veterinarian clinics for preventing tick bite in dogs (S. Doggett, personal communication). As over 1000 companion animals every year are affected by tick paralysis [15], such devices, if ineffective, could provide the owner with a false sense of security, thereby risking the pets’ health when taken into tick-prone areas.

Ultrasonic sound has a frequency of more than 20 kHz, which many arthropods can detect [16]. Some arthropods have been reported to be repelled by ultrasonic frequencies in the range of 25 to 65 kHz [17]. It is thought that this frequency creates stress in the nervous system of arthropods, causing them to move away from the source [16]. However, to date, there have been no published reports that have demonstrated that ultrasonic sound effectively repelled any arthropod pest.

Huang et al. found that commercial ultrasound devices failed to repel ants in both laboratory and field trials [18]. A lack of repellency was also reported with cockroaches [19,20], mosquitoes [21], fleas [22,23] and the common bed bug, *Cimex lectularius* L. [24]. For ticks, there are very few reports that have evaluated the effectiveness of ultrasonic repellents and the one study undertaken to date found such devices ineffective [25]. In spite of the lack of evidence for the efficacy of ultrasonic devices in repelling ticks, as noted above, they are still widely available. In light of their availability and a lack of research on these devices against Australian ticks, the aim of this study was to evaluate the effectiveness of a range of commercially available ultrasonic pest repellent devices to repel *I. holocyclus*.

## 2. Materials and Methods

### 2.1. Ticks

Adult female *I. holocyclus* were used in the study as it is this stage that poses the greatest health risk to humans and pets. Ticks were collected by flagging in natural habitats in Irrawong Reserve, New South Wales, Australia (33°41′ S, 151°17′ E), over October to November 2019. Collected ticks were maintained in 20 mL sample jars containing 2 cm of Plaster of Paris in the base of each jar. The lid of each jar was modified such that a 1 cm diameter hole was removed and replaced with chiffon fabric that was glued to the lid to allow airflow into the jar. Three drops of distilled water were supplemented into each jar on a regular basis (every few days) to maintain humidity. The ticks were held for 2–4 days under laboratory conditions at temperatures 25–27 °C with 70–80% relative humidity (RH) and 12:12 h (light: dark) prior to testing. As insufficient ticks were collected for the study from the field site, additional, female *I. holocyclus* were purchased from Australian Veterinary Serum Laboratories, Lismore, NSW. These purchased ticks were collected from undisclosed field sites by individuals on behalf of the company and maintained in the laboratory as above.

### 2.2. Ultrasonic Pest Repellent Devices

Nine commercial ultrasonic pest repellent devices were evaluated (Table 1). They were purchased online through eBay and operated according to manufacturer’s instructions. The operating frequency for each device, as provided by manufacturers, is listed in Table 1. Most devices were portable and battery-operated, except for devices H and I that required AC power and were designed for household use. For the portable ultrasonic devices, devices D and G (according to the manufacturers’ claims) were designed for placement around the neck of pets for protection against fleas, ticks, and mosquitoes. Both of these did not list the operating frequency used. From the nine devices, there were five (B, E, F, H, and I) that the instructions claimed that they repelled mosquitoes and other pests, but did not specifically mention ticks. However, these were included in the study to comprehensively test a range of devices that cover different operating frequencies.

### 2.3. Test Chambers

The testing of the ultrasonic devices was based on the procedures of Huang et al. [18] and Yturralde and Hofstetter [24]. Two test arenas were constructed, one was used as the study group (with an ultrasonic device) and the other as a control (no ultrasonic device). Each test arena comprised of two chambers (each 30 × 30 × 30 cm^3^) constructed from transparent Perspex and connected towards the bottom by a cardboard corridor (3 × 10 × 10 cm^3^) that allowed for the ticks to escape (Figure 1). It was possible to close the corridor off with a gate to prevent ticks from entering the escape chamber (‘a’ in Figure 1). The ultrasonic devices were tested separately and each was suspended into the ‘test’ chamber (‘b’ in Figure 1) at the height of 20 cm above the floor of the chamber using a ring stand and 40 cm away from the escape chamber. Twenty-four hours before the test, the interior of each chamber was lined with Fluon (polytetrafluoroethylene suspension; BioQuip, Rancho Dominguez, CA, USA) along the top 10 cm to prevent ticks from escaping. After the ultrasonic device was turned on, the lid of each chamber was closed to reduce noise contamination between the chambers.

### 2.4. Testing Procedures

In each trial, twenty *I. holocyclus* ticks were introduced into the test chamber and allowed to acclimate to the test conditions for 30 min with the corridor closed. After the acclimation period, the gate was opened and the ultrasonic device in the test chamber turned on for 60 min. After 60 min, the gate was closed and the number of ticks in each chamber and inside the corridor counted. There were 10 replicates for each device and ten replicates for the controls. Test conditions were 25–27 °C with 70–80% RH. Humidity in test chambers was maintained through the trials being conducted in an insectary that has an automatic humidifier controller installed. To reduce any inherent position bias caused by extraneous factors (for example, light from the window), the test arenas were rotated 180° between each replicate.

### 2.5. Data Analysis

The percentages of ticks that escaped from the ultrasonic devices was calculated by the number of ticks found in both the escape chamber and the corridor multiplied by 100, and divided by the number of tested ticks (10 replicates: A total of 200 ticks for devices A-H and 60 ticks for device I). The test results were then adjusted from the corresponding controls using Abbott’s formula [26]. Data on the number of escaped ticks between the test and control arenas of each device were compared using the Mann–Whitney U test. The null hypothesis (H0) assumes that there were no differences between the tick escape rate in the test and control arenas, while the alternative hypothesis (HA) assumes that they were different. The overall relationship between the number of escaped ticks between the test and control arenas was explored using the Generalized Linear Mixed Models (GLMMs) fitted by the restricted maximum likelihood (REML) approach. The number of escaped ticks was treated as the dependent variable, while trial arenas as the factors, and the devices as the cluster variable. The trial arenas were defined as the fixed effect and device ID as a random effect. Statistical significance for all tests was set at 5% (*p* < 0.05). Data were analyzed using Jamovi version 1.2 (the Jamovi Project, www.jamovi.org, accessed on 20 January 2020) and SPSS Statistics version 22 (IBM Corp., Armonk, NY, USA).

## 3. Results

The results of the tests using the nine ultrasonic devices are summarized in Table 2 and graphically depicted in Figure 2. After each ultrasonic device was turned on in the test chamber for 60 min, a number of ticks moved to the escape chamber, with less than 4% being found in the corridor between the two chambers. However, most of the ticks remained scattered in the test chamber. Similar results were obtained with the control arena. From the trials with the ultrasonic devices, the highest percentage of escaped ticks, when corrected by Abbott’s formula from the controls, was with device D (19.5%, n = 51, Table 2) followed by F (16.8%, n = 47), I (12.8%, n = 12), B (11.9%, n = 44), G (11.6%, n = 39), A (8.1%, n = 41), C (7.4%, n = 38), E (5.5%, n = 29) and H (4.5%, n = 31). Interestingly, in all the trials, the number of ticks that escaped in the test arena was always greater than the control, albeit not always statistically different (Table 2 and Figure 2). The GLMM analyses revealed that there was a trend of greater repellency in all devices compared to the controls (*p* < 0.001) (Table 3). This repellent effect was only significant for devices B, D, and F (*p* = 0.026; 0.004; 0.014, respectively). Throughout the whole experiments, no mortality was observed.

## 4. Discussion

The present study performed the testing of the repellent efficiency of various ultrasonic devices over one hour based on the experimental design of Yturralde and Hofstetter [24]. These researchers turned on the ultrasonic devices immediately for 30 min after 10 bed bugs had been released into the test arenas. For our study, the size of the test arenas (30 × 30 × 30 cm^3^) was comparable to their protocols (29.85 cm in diameter, 36.8 cm in height), although our study employed a greater number of test specimens, namely 20 ticks per trial. Based on the size of our test arenas and number of tested ticks, we adjusted the exposure time to one hour. This amount of time is considered adequate as a device that is claiming to be repellent by the manufacturer should provide complete protection within this time frame.

The overall results of the statistical analyses indicated that some of the tested devices tended to repel ticks compared to the control groups, although the level of repellency observed was very low (less than 20%). As more than 80% of the ticks were not repelled within the confined area, this level of repellency is clearly insufficient to provide adequate protection from a potential tick bite. Our results are comparable to previous reports whereby ultrasonic devices were found unable to repel pests [18,19,20,21,22,23,24,25,27,28]. Some studies have focused more on the repellency rate data obtained by statistical analysis rather than the behavior of arthropods after detecting sound waves from an ultrasonic device. In our investigation, ultrasound had some effect on the tick movement. A number of ticks stopped or moved slowly after stimulation of ultrasound-producing devices, which is potentially significant in terms of host protection. In such a scenario, sluggish moving ticks or those that ceased movement were less likely to attach to the host but would after a device was disconnected.

The behavior of ticks in the test arena was observed during trials. After ticks were released in the test chamber and allowed to acclimate for 30 min, they displayed normal motility throughout the chamber. Some ticks moved notably slowly after the ultrasonic devices were turned on. The results demonstrated that the frequency of sound did affect tick behavior and movement as a statistically significant repellent effect was recorded in devices B, D, and F, but very low in other devices. Consistent with some previous studies on ultrasonic pest controllers, Yturralde and Hofstetter reported that ultrasonic devices reduced the movement of *C. lectularius* [24]. In contrast, flea behavior was unaffected by ultrasonic devices [28]. Likewise, Brown and Lewis found that ultrasonic devices did not affect the behavior of the tick *Rhipicephalus simus*, even when the tick was within 1 cm of the device [25]. These authors also found that ticks under exposure to the ultrasonic frequency still responded to the external stimuli of a gentle exhalation from the experimenter.

As noted in Methods, ultrasonic devices tested were all commercially available and sound frequencies emitted by each as per manufacturers’ claims were slightly dissimilar (albeit with considerable overlap). Results showed that all devices poorly repelled *I. holocyclus* even though there were three significant results. Some devices were not marketed specifically as tick repellents, such as devices B and F. The product manual of device B claimed that it emits very fast and powerful 5–20 kHz multi-frequency sound waves to repel annoying mosquitoes, while device F is claimed that it can remove mosquitoes and is also effective in repelling pests, such as cockroaches, flies, and even rodents. However, the highest escape percentage of *I. holocyclus* was only 19.5% of ticks being repelled. It would be expected that repellency would even be lower in an open field situation, where sound waves would be more dispersed. Several studies have been conducted to test the performance of ultrasonic devices in the field. Ultrasonic pet-collar devices were ineffective in reducing flea numbers on cats [29,30]. Additionally, Schein et al. showed no difference between the numbers of fleas and ticks initially placed on dogs with ultrasonic pet-collars and on control dogs, even after 14 days of device operation [29]. For device I in our investigations, which claimed to be capable of repelling a variety of arthropods and animals, only three replicates were performed as the device failed during the experiment.

## 5. Conclusions

In summary, all nine commercial ultrasonic sound pest repellent devices tested in this study demonstrated low-level repellency against *I. holocyclus* in the confined test arena. However, the small amount of repellency observed would be insufficient to offer adequate protection against tick bites. Thus ultrasonic devices are not recommended for use in the prevention of tick bites from *I. holocyclus.*

## Figures and Tables

**Figure 1 insects-12-00400-f001:**
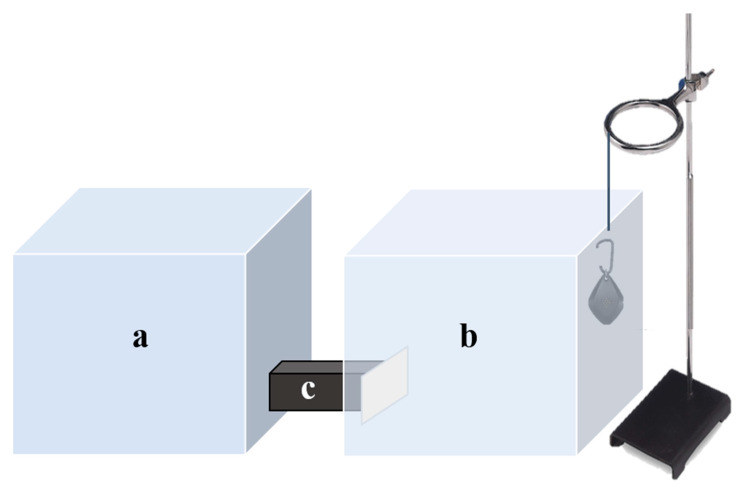
The test chambers used in the trials. The escape chamber (**a**) connected the test chamber (**b**) by a cardboard corridor (**c**). For the treatment, ultrasonic device was suspended at a height of 20 cm within test chambers by using a ring stand and the other as a control (no ultrasonic device).

**Figure 2 insects-12-00400-f002:**
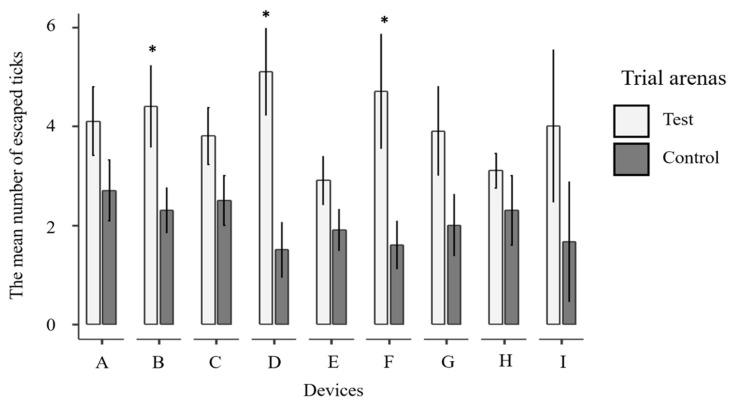
The mean number of ticks escaping (into chamber a) in the ultrasonic treatment and control (no sound) arenas. Error bars represent standard errors. An asterisk (*) above each pair of bars indicates a statistically significant difference.

**Table 1 insects-12-00400-t001:** The nine ultrasonic devices used, along with their frequency (in kilohertz), as stated by the manufacturers.

Devices	Manufacturer	Ultrasonic Frequency (kHz)
A. TICKLESS ^®^ PET Ultrasonic Tick and Flea Repeller for Pet	ProtectONE Ltd., Budapest, Hungary	40
B. MOZZIGEAR^TM^ Portable Ultrasonic Mosquito Repeller	Intelligent Health Systems, Guangdong, China	5–20
C. TICKLESS ^®^ HORSE Ultrasonic Tick and Flea Repeller for Horses	ProtectONE Ltd., Budapest, Hungary	40
D. Pet’s Pest Repeller	www.Petshopboyz.com.au, accessed on 28 September 2020, Sydney, Australia	n/a
E. L1-118 Portable Electronic Insect Repellent	Shenzhen Dowdon Tech Co., Ltd., Guangdong, China	9–21
F. Portable Smart Pest Repeller	Shenzhen Dowdon Tech Co., Ltd., Guangdong, China	13–75
G. CSB24 Ultrasound device against ticks and fleas	Intelligent Health Systems, Guangdong, China	n/a
H. ELECTRONIC HELMINTHES MACHINE	Hunan Goldenserise Tech Co., Ltd., Hunan, China	22–65
I. ULTRASONIC PEST REPELLER Pest Reject	Hunan Goldenserise Tech Co., Ltd., Hunan, China	50–60

n/a = not provided.

**Table 2 insects-12-00400-t002:** The number (and percentage) of escaped (chamber a) and non-escaped (chamber b) ticks after one hour of operating the ultrasonic device and the control (no ultrasonic device).

Devices	Trials	No. of Tick (%)		*p*-Value
		Escaped †	Non-Escaped	
A	Test	41 (8.1 ‡)	159 (79.5)	
	Control	27 (13.5)	173 (86.5)	0.100
B	Test	44 (11.9 ‡)	156 (78)	
	Control	23 (11.5)	177 (88.5)	0.026 *
C	Test	38 (7.4 ‡)	162 (81)	
	Control	25 (12.5)	175 (87.5)	0.071
D	Test	51 (19.5 ‡)	149 (74.5)	
	Control	15 (7.5)	185 (92.5)	0.004 *
E	Test	29 (5.5 ‡)	171 (85.5)	
	Control	19 (9.5)	181 (90.5)	0.121
F	Test	47 (16.8 ‡)	153 (76.5)	
	Control	16 (8)	184 (92)	0.014 *
G	Test	39 (10.6 ‡)	161 (80.5)	
	Control	20 (10)	180 (90)	0.065
H	Test	31 (4.5 ‡)	169 (84.5)	
	Control	23 (11.5)	177 (88.5)	0.055
I	Test	12 (12.8 ‡)	48 (80)	
	Control	5 (8.3)	55 (91.7)	0.184

* A significant difference (*p* < 0.05) in the number of ticks that escaped between the test and control arenas. † The number and percentage of escaped ticks in both the escape chamber and corridor. ‡ The percentage of ticks escaped in the tests was corrected from the escaping ticks in the controls using Abbott’s formula.

**Table 3 insects-12-00400-t003:** Parameters estimated by the best fit generalized linear mixed model (GLMM).

Effect	Estimate	SE	95% Confidence Interval		df	t	*p*-Value
			Lower	Upper			
(Intercept)	3.04	0.162	2.73	3.36	164	18.81	<0.001
Control-Test	−1.92	0.324	−2.55	−1.28	164	−5.92	<0.001

Significance level set at *p* < 0.05.

## Data Availability

All relevant data are included in the article.

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
