# Peer review of "The Efficacy of Ultrasonic Pest Repellent Devices against the Australian Paralysis Tick, Ixodes holocyclus (Acari: Ixodidae)"

_insects, 2021, doi:10.3390/insects12050400_

Round 1

Reviewer 1 Report

The paper is very good. The authors raised an interesting topic, but at the same time it could raise some controversy.

The manuscript is well written. The authors adapted to the publisher's requirements, however I suggest the authors read the text again and pay attention to the necessity of italicizing the Latin names.

My main comment is that the authors used both field-collected and purchased ticks (probably from a laboratory colony). This could have affected the activity of the ticks. Ticks from these two groups were certainly of different physiological ages and were characterized by a different content of, for example, fat bodies, water - components that significantly affect the activity of ticks.

In the future, the authors could extend their research with an animal model.

Nevertheless, I suggest acceptance the manuscript for publication.

Reviewer 2 Report

Review of the article „The efficacy of ultrasonic pest repellent devices against the 2Australian Paralysis tick, Ixodes holocyclus (Acari: Ixodidae)” by A. Panthawong, S.L. Doggett, and Theeraphap Chareonviriyaphap

I express my congratulations to the Authors on their results. The presented results are important, as they not only enrich our knowledge but show the real effectiveness of devices advertised as repelling ticks and recommended by manufacturers for tick bite prevention.

The study was carried out correctly and with great care for the reliability of laboratory tests, and its results were critically analyzed based on the available literature on the subject. Undoubtedly, this original study will arouse readers' interest and will influence tick bite preventive behaviors. Therefore, I strongly support the Authors' efforts to publish the study in Insects.

Minor comments:

  1. Introduction

lines 37-40, I suggest re-editing the sentence as follows:

Not only is the species capable of transmitting rickettsia such as Rickettsia australis Philip (etiological agent of Queensland tick typhus) [3], the bite of the tick can cause potentially life-threatening toxicosis, tick paralysis and induce a range of allergic reactions such as α-Gal syndrome (mammalian meat allergy) [5,6]

line 38, Rickettsia australis should be written in italics

line 48 Please change „applied to skin” to „applied to clothing and skin”

lines 51, 53, 111 etc. For the type of citations used by the Insects journal, it is not necessary to provide publication dates in parentheses, the number of the cited item in square brackets is enough so I suggest you delete the dates as below throughout the manuscript:

Fortunately, Sukkanon et al. recently tested a range of topical and spatial repellents and made recommendations on which were the most effective products at repelling I. holocyclus[13]. Even more recently, Panthawong et al. investigated the use of permethrin-impregnated clothing for repelling I. holocyclus [14]. lines 50-54

line 58 I suggest changing (S. Dogget 2020, unpublished data) to (S. Dogget, personal communication)

  1. Materials and Methods

2.1. Ticks

line 84 change mL to ml

2.3. Test chambers

line 112 Please change „the treatment” to „the study group”

2.4. Testing procedures

line 135 please specify how 70-80% RH in test chambers was maintained

line 137, change 180o to 180°

  1. Discussion

line 213 please change „movement” to „motility”

line 220 change Rhipicephalis to Rhipicephalus

Take care and stay safe

Reviewer

Reviewer 3 Report

The manuscript “The efficacy of ultrasonic pest repellent devices against the Australian Paralysis Tick, Ixodes holocyclus (Acari: Ixodidae)” is being considered in the journal Insects. This manuscript investigated I. holocyclus. A controversial area of pest control is using ultrasonic sound to deter pests with previous research showing that that the devices are not effective. Nine ultrasonic repellent devices were evaluated. Ticks were exposed to sounds for one hour, aligning to the manufacturer’s instructions. Some of the tested devices would repel ticks when compared to the control; however, the level of repellency was low. The conclusion of the paper is that low-level repellency was observed with I. holocyclus in a confined arena and likely will not be a good way to repel ticks.

Overall, the studies are straight forward study, and the results are important in showing the lack of efficacy of ultrasonic sound to control insect pests, including ticks.  The conclusion of the paper are supported by the results and the manuscript would likely be of interest to the readers of this publication.  I have no further suggestions requiring this manuscript. 
